# Mothers' acceptability of using novel technology with video and audio recording during newborn resuscitation: A cross-sectional survey

So Yeon Joyce Kong[1], Ankit Acharya[2], Omkar Basnet[2], Solveig Haukås Haaland[1], Rejina Gurung[2,3], Øystein Gomo[1], Fredrik Ahlsson[3], Øyvind Meinich-Bache[1,4], Anna Axelin[5], Yuba Nidhi Basula[6], Sunil Mani Pokharel[6], Hira Subedi[6], Helge Myklebust[1‡], Ashish KC[7‡*]

1 Laerdal Medical, Stavanger, Norway, 2 Golden Community, Chakupat, Lalitpur, Nepal, 3 Department of Women's and Children's Health, Uppsala University, Uppsala, Sweden, 4 University of Stavanger, Stavanger, Norway, 5 University of Turku, Turku, Finland, 6 Bharatpur Hospital, Chitwan, Nepal, 7 School of Public Health and Community Medicine, Institute of Medicine, University of Gothenburg, Gothenburg, Sweden

‡ These authors are joint senior authors on this work.
* ashish.k.c@gu.se

**Data Availability Statement:** We have now deposited the data in cvs format.

## Abstract

### Objective

This study aims to assess the acceptability of a novel technology, MAchine Learning Application (MALA), among the mothers of newborns who required resuscitation.

### Setting

This study took place at Bharatpur Hospital, which is the second-largest public referral hospital with 13 000 deliveries per year in Nepal.

### Design

This is a cross-sectional survey.

### Data collection and analysis

Data collection took place from January 21 to February 13, 2022. Self-administered questionnaires on acceptability (ranged 1–5 scale) were collected from participating mothers. The acceptability of the MALA system, which included video and audio recordings of the newborn resuscitation, was examined among mothers according to their age, parity, education level and technology use status using a stratified analysis.

### Results

The median age of 21 mothers who completed the survey was 25 years (range 18–37). Among them, 11 mothers (52.4%) completed their bachelor's or master's level of education,

**Funding:** This pilot study was funded from a research funding by Strategic Research, Laerdal Medical, Stavanger, Norway. None of them were involved in any direct participation in the execution, administration, or data collection process of the survey.

**Competing interests:** Co-authors SYJK, SHH, ØG, ØMB, and HM are employees of Laerdal Medical.

13 (61.9%) delivered first child, 14 (66.7%) owned a computer and 16 (76.2%) carried a smartphone. Overall acceptability was high that all participating mothers positively perceived the novel technology with video and audio recordings of the infant's care during resuscitation. There was no statistical difference in mothers' acceptability of MALA system, when stratified by mothers' age, parity, or technology usage (p>0.05). When the acceptability of the technology was stratified by mothers' education level (up to higher secondary level vs. bachelor's level or higher), mothers with Bachelor's degree or higher more strongly felt that they were comfortable with the infant's care being video recorded (p = 0.026) and someone using a tablet when observing the infant's care (p = 0.046). Compared with those without a computer (n = 7), mothers who had a computer at home (n = 14) more strongly agreed that they were comfortable with someone observing the resuscitation activity of their newborns (71.4% vs. 14.3%) (p = 0.024).

## Conclusion

The novel technology using video and audio recordings for newborn resuscitation was accepted by mothers in this study. Its application has the potential to improve resuscitation quality in low-and-middle income settings, given proper informed consent and data protection measures are in place.

### Author summary

Effective newborn resuscitation reduces intrapartum stillbirth and newborn death, yet health care providers cannot provide high quality care despite training and continuous quality improvement. We developed a novel technology which provides real time guidance on time since birth in a monitor, as well as records audio and video of resuscitation, this technology has been shown to be acceptable to health care providers. In this study, we assessed how acceptable was this novel technology for resuscitation by mothers through a self-administered questionnaire to 21 participants. Overall acceptability was high that all participating mothers positively perceived the novel technology with video and audio recordings of the infant's care during resuscitation. There was no difference in mothers' acceptability to the technology, when stratified by mothers' age, parity, or technology usage. Compared with those without a computer, mothers who had a computer at home more strongly agreed that they were comfortable with someone observing the resuscitation activity of their newborns.

## Introduction

Globally 2.4 million newborns die out of 140 million live births every year, despite the efforts of Millennium Development Goals (MDGs) and Sustainable Development Goals (SDGs) of the past two decades [1]. Eighty-four percent of the countries at risk of failing to meet the newborn mortality target of the SDGs are low- and middle-income countries (LMICs) [1]. The newborn mortality rate is highest in Sub-Saharan Africa with 27 deaths per 1000 live births, followed by South Asia with 23 deaths [1]. The high newborn mortality rate in these regions is primarily due to inadequate access to effective, high-quality care for infants born prematurely or with low birth weight, as well as complications from asphyxia, sepsis, infections and intrapartum injuries [1–5]. Each year, more than 10 million newborn who do not breathe

immediately at birth require lifesaving interventions such as newborn resuscitation [6–9]. Resuscitation has been prioritized as a cost-effective evidence-based solution for preventing newborn without breathing from dying [10–14]. Since 2010, Helping Babies Breathe (HBB), a newborn resuscitation program designed specifically for resource-constrained settings, has been implemented in over 80 countries worldwide and shown a significant improvement in health care providers' (HCPs) knowledge, skills, and competency as well as neonatal mortality [15–18]. However, it has also been demonstrated that newborn resuscitation skills deteriorate rapidly over time [19–21]. Moreover, low concordance between knowledge and skills and sub-standardized levels of knowledge and skills of newborn resuscitation have been identified as well [22]. As newborn resuscitation is effective only when HCPs have sufficient skills and knowledge, various methods have been introduced and used to improve clinical performance for resuscitation, including video recording of resuscitations.

The use of video recording of resuscitations has been a valuable tool in multiple aspects of the resuscitation process from training to quality assurance. In particular, reviewing video recording of newborn resuscitation procedures have shown to be effective intervention for improving skills and clinical outcomes [23–26]. However, since review of resuscitation proce-dure is performed after, not during, the intervention, HCPs still need to depend upon their own cognitive skills and memory during resuscitation, which may result in suboptimal resusci-tation with delays in initiating ventilation. To mitigate this, we are currently in the process of developing an automated guidance to HCPs during neonatal resuscitation using a deep learn-ing model called MALA (MAchine Learning Application), which is based on automatic video analysis and activity recognition [27].

MALA will be a tablet-based software application that provides real-time guidance to HCPs for next step of resuscitation through visual display and audio prompts during resuscitation. The real-time guidance will be based on the automatic analysis of the event from visual and audio activities recorded by a tablet; therefore, the development of MALA will require a large amount of video and audio recordings to train and deploy machine learning. As video and audio activities are the main components of MALA application, feasibility and acceptability of video and audio recordings to intervention deliverers (HCPs) and recipients (parents/new-borns) are crucial factor to consider in the development, evaluation and implementation phases of MALA [28]. Currently, a pre-version of MALA application with video and audio recordings has been developed to guide the research team on further development of MALA application, regarding feasibility and acceptability of video and audio recordings. We previously assessed the usability, feasibility, and acceptability of this pre-version of MALA among HCPs and demon-strated reasonable usability, feasibility and acceptability of this novel technology with visual guidance on elapsed time and with video and audio recording during newborn resuscitation [29]. As MALA will process personal data, which may only be processed with appropriate con-sent by the data subjects, acceptability of MALA among mothers is very important. In this study, we further assessed the acceptability of MALA for mothers of resuscitated newborns.

## Methods

### Study design

This is a cross-sectional survey assessing the acceptability of the novel technology among mothers of newborns requiring resuscitation.

### Study setting

This study was conducted between January 21 and February 13, 2022 in Bharatpur Hospital, Chitwan District, Nepal, which is the second-largest government and a tertiary referral

hospital. There are over 13,000 annual deliveries in Bharatpur Hospital. The delivery unit has in total 21 beds (3 for admissions, 15 for labour management, and 3 for delivery care) and newborn resuscitation corners. The stillbirth and neonatal mortality rates are estimated at 11 and 2 per 1000 live births respectively, for 1 year period from March 2019 to March 2020. There are about 30 HCPs working in the maternity ward and all of them received HBB training.

## Study participants

All women giving birth at Bharatpur Hospital during the study period whose newborns underwent resuscitation were eligible to participate in the survey. Exclusion criteria for women included age under 18 years old. All the mothers, who were eligible to the study, were approached by the research team. Research team informed the participants about the study, including the video and audio recording of the newborn resuscitation using tablet-based application. The participants were informed on the rationale on having the video and audio recording to improve the quality of resuscitation care as well as future improvement in the technology. Those mothers who agreed to be part of the study on using the video recording for resuscitation were included in the study and thus the technology was used among the consented mothers. Further, after the video recording of the resuscitation events, mothers were again approached whether they consent the video recording to be used for research purpose.

## Intervention (pre-version MALA application)

Details of the pre-version of MALA application has been previously described [29]. The novel technology consisted of an infant warmer (Phoenix Medical Systems, Chennai, India) that is equipped with a tablet, which recorded video and audio of resuscitation activities and provided visual guidance in elapsed time since birth. The infant warmer is also equipped with a newborn heart rate meter called Neobeat (Laerdal Medical, Stavanger, Norway), a manual suction device, a bag-and-mask resuscitator with Positive end-expiratory pressure(upright with PEEP) and newborn bag-mask (Laerdal Medical) (Fig 1A). Newborn status and treatment were manually annotated using Liveborn app (Laerdal Global Health, Stavanger, Norway). Liveborn app is a mobile application, used for research on newborn resuscitation, where an observer can document the timing of birth and resuscitation activities such as dry/stimulation, skin-to-skin, clamp cord, suction, and ventilation by using push buttons [30]. When 'baby born' button is pressed on Liveborn application at the time of birth, video recording is automatically started on the tablet mounted onto the infant warmer and captures the newborn and HCPs' hands (Fig 1B). Newborn resuscitation events were observed and annotated in the Liveborn application, and the newborn heart rate (HR) was streamed from NeoBeat to the Liveborn application. When no further resuscitative care was provided by HCPs, observation was ended in the Liveborn application, and the video recording was automatically stopped. Then the recorded video (Fig 1B), annotations of resuscitation activities from Liveborn app, newborn heart rate and accelerometer signal from NeoBeat (Fig 2) were uploaded to a highly secured data storage system using Microsoft Azure. These different types of resuscitation activity data will be extracted and used to further analyse quality of resuscitation activities. If a newborn did not need any resuscitative care after birth, the already initiated observation was cancelled in Liveborn application and the video recording stopped automatically and was deleted from the data system.

## Development of the survey tool

The questionnaire included self-reported demographic characteristics (mother's age, parity, and education), technology usage (having a computer at home, having a smart phone, using apps, and posting pictures in social medias), and questions related to the novel technology

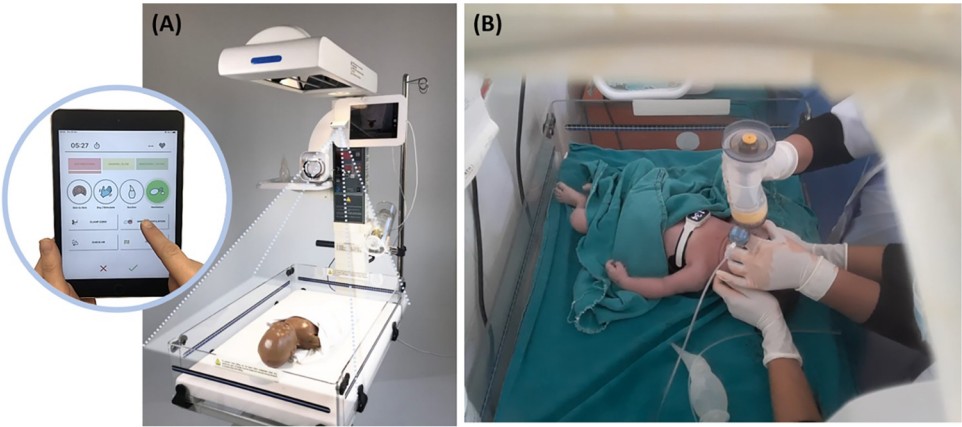

**Fig 1. The MALA system (A) and real capture from a recorded video during newborn resuscitation of a newborn with Neobeat (newborn heart rate meter that provides real-time heart rate and motion data) and Upright bag with PEEP at Bharatpur hospital (B).** The MALA system is equipped with infant warmer, a tablet computer with a camera for sound and video recording, Liveborn app, NeoBeat, a manual suction device, and Upright bag with PEEP. Air tube connects PEEP value to the microphone on the tablet for better sound recording of ventilation quality. At the time of birth when 'baby born' button is pressed on Liveborn app, video recording of newborn resuscitation is automatically started on the tablet mounted onto the infant warmer, which records the area indicated by dotted lines thus captures only the newborn and the health care providers' hands. The camera captures only newborn and the health care providers' hands. Video recordings of newborns without any resuscitative care after birth are automatically deleted from the system.

acceptability and feasibility. The reliability of questionnaire was also assessed using the Cronbach's Alpha for all seven items, which was 0.859 (S1 Table).

The self-administered acceptability questionnaire was composed of 7 questions assessing perceived acceptability of the novel technology using a Likert scale of 1 to 5, where 1 represents strongly disagree and 5 represents strongly agree. Acceptability-related questions were developed by the research team based on the parents acceptability questionnaire for bedside

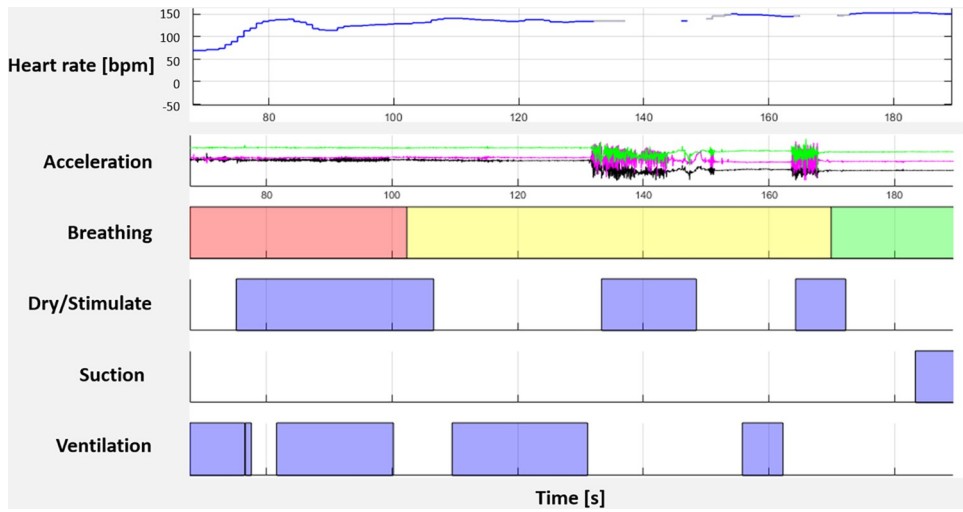

**Fig 2. Data generated based on the newborn heart rate and accelerometer signals from Neobeat and annotations of resuscitation activities (breathing, dry/stimulate, suction, and ventilation) from Liveborn app.** Heart rate (in beats per minute) is recorded by Neobeat and live streamed to Liveborn app. Accelerometer inside NeoBeat measures movement in 3-axis. Data collectors/observers annotate resuscitation activities in Liveborn app. For Breathing activity, red indicates "not breathing", yellow indicates "gasping/slow breathing", and green indicates "breathing well".

resuscitation with intact umbilical cord intervention developed by Katheria *et al.* [31] and after several rounds of discussions among the research team. The final questionnaire was translated into Nepali language.

## Data collection

Data collection was conducted from January 21 to February 13, 2022. All participating mothers, except one mother who was illiterate, completed self-administered survey. For the illiterate mother, survey was completed through an interview. Data collectors collected the questionnaire from the participating mothers after resuscitation, which were then entered into the database system for further analysis.

## Data analysis

The categorical variables were expressed as frequencies (percentage) and continuous variables expressed as the median and range. Stratified analysis was conducted on mothers' acceptability of the technology by mothers' age, parity, education level, and technology use (computer, smartphone, and social media). For stratified analysis, p-values were calculated based on Fisher's Exact test. A p-value less than 0.05 was considered statistically significant. Data analysis was performed using SPSS Software (IBM SPSS Statistics for Windows, V.23.0).

## Ethical consideration

The principle of informed consent has been adapted [32] and good clinical practice (GCP) guidelines from International Conference on Harmonization was implemented [33]. After a thorough explanation of the procedures, including the risks and benefits, before the admission to labour room, the mothers' consent for participation in the study including their permission for newborn resuscitation and video-filming during clinical procedures, was obtained [34].

## Results

Among the 243 deliveries that took place during the study period, 240 mothers consented to participate in the survey. Among 32 newborns who required resuscitation, 24 had difficulty breathing and were taken to the resuscitation table for further intervention using bag-and-mask or suction-and-stimulation. All resuscitated newborns survived and were safely returned back to the mother. All 24 mothers consented to participate, and 21 completed the survey. The Liveborn app was used to annotate the treatment of all 21 newborns who were brought to the infant warmer for resuscitation. We compared the socio-demographic characteristics of women who did not consent with women (n = 3) who consented (n = 21). There was no difference in age, parity, ethnicity but education level was higher among the women who did not consent (S2 Table).

Along with suction and stimulation, bag and mask ventilation were required for 20 newborns, and only suction and stimulation were required for 1. All but one of those who needed a bag and mask had a Neobeat, heart rate meter attached to them, among which 15 newborns had it linked with the liveborn app and HR data was continuously streamed to the app (Table 1).

The median age (range) of the participating mothers was 25 years (18–37). Thirteen of them (61.9%) had delivered their first child, six (28.6%) had delivered their second, and two (9.5%) had delivered their third child. One mother (4.7%) was illiterate, 3 mothers (14.3%) had completed their secondary education, 6 mothers (28.6%) had completed their higher secondary education, and 11 mothers (52.4%) had completed their bachelor's or master's degrees.

**Table 1. Characteristics of the participating mothers.**

| Variables | Number (%) |
|---|---|
|  | (Total = 21) |
| *Demographic factors* |  |
| Age, years (median, range) | 25 (18–37) |
| Parity |  |
| First baby (1) | 13 (61.9) |
| Second baby (2) | 6 (28.6) |
| More than 2 babies (3≥) | 2 (9.5) |
| Education |  |
| Illiterate (unable to read and write) | 1 (4.7) |
| Up to primary level (Up to grade 5) | 0 |
| Up to secondary level (Up to grade 10) | 3 (14.3) |
| Up to higher secondary level (SEE above) | 6 (28.6) |
| Bachelor's level | 6 (28.6) |
| Master's level | 5 (23.8) |
| Masters and above | 0 |
| *Technology Usage* |  |
| I have a computer at home, Yes | 14 (66.7) |
| I have a smart phone, Yes | 16 (76.2) |
| *I use apps in my smart phone** |  |
| Never | 1 (6.2) |
| Monthly | 0 |
| Weekly | 1 (6.2) |
| Daily | 14 (87.6) |
| I post pictures in social media |  |
| Never | 7 (33.3) |
| Monthly | 7 (33.3) |
| Weekly | 1 (4.8) |
| Daily | 6 (28.6) |

*Among those who have a smart phone (n = 16)

Fourteen mothers (66.7%) owned a computer at home. Sixteen mothers (76.2%) carried a smartphone device with them, of them 14 (87.6%) used mobile application on daily basis. Seven mothers (33.3%) never posted photos on social media while 6 mothers (28.6%) posted photos on a daily basis.

Fig 3 describes the result of maternal acceptability of the MALA system. Overall, no mothers responded 'disagree' or 'strongly disagree' on any of the acceptability-related questions. In terms of whether mothers were comfortable with videorecording of the newborn care, 11 mothers (52.4%) agreed, 8 mothers (38.1%) strongly agreed and remaining 2 (9.5%) had neutral agreement. Ten mothers each (47.6%) agreed and strongly agreed that they were comfortable with someone using a tablet while observing their newborn's care and the remaining mother (4.8%) shared neutral agreement. Ten mothers (47.6%) agreed and 11 (52.4%) strongly agreed that they were comfortable with someone observing the newborn resuscitation activity of their baby. All mothers trusted that the information of their newborn will be kept strictly confidential (61.9% for agree and 38.1% for strongly agree) within the MALA system. In terms of the use of video and audio recording during resuscitation will neither cause harm nor will it compromise the care of their baby in the hospital, 11 mothers (52.4%) agreed and 10 (42.9%)

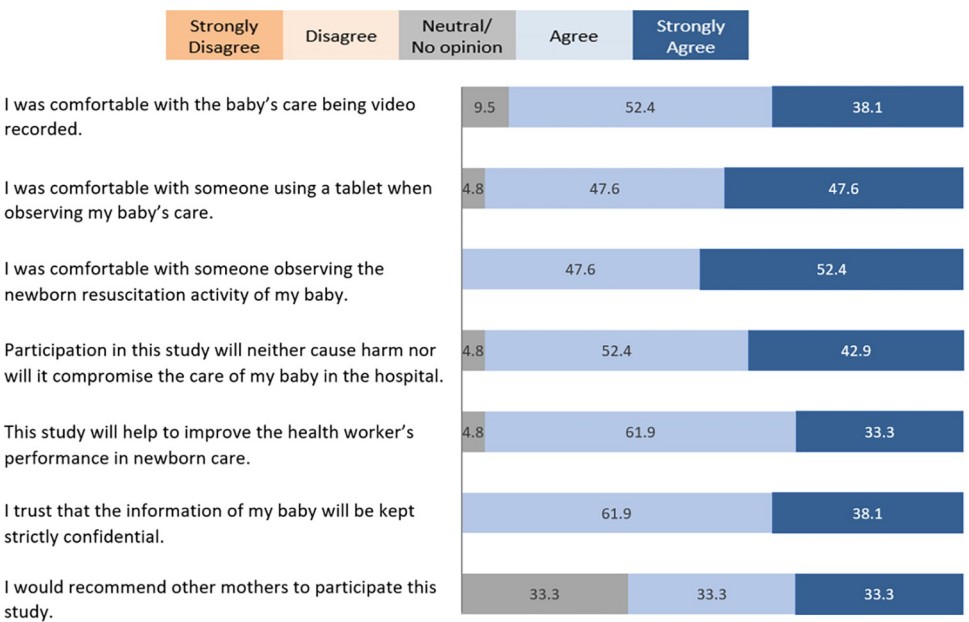

**Fig 3. Mothers' acceptability of the MALA system in percentage (N = 21).** When the acceptability of the MALA system was stratified by mothers' median age (≤25 years vs. >25 years) and parity (primiparity vs. multiparity), there was no statistical difference in mothers' acceptability of MALA system (p>0.05 for all) (S3 and S4 Tables). However, when the acceptability of the technology was stratified by mothers' educational level (up to higher secondary level vs. bachelor's level or higher), mothers with Bachelor's degree or higher more often strongly agreed that they were comfortable with the baby's care being video recorded (p = 0.026) and someone using a tablet when observing the baby's care (0.046) (S5 Table). Compared with those without a computer (n = 7), mothers who had a computer at home (n = 14) more often strongly agreed that they were comfortable with someone observing the newborn resuscitation activity of their baby (71.4% vs. 14.3%) with a statistical significance (p = 0.024) (S6 Table). There was no statistically significant difference in the acceptability of the MALA system in terms of other technology usage (having a smart phone, using smart phone app, using social media).

strongly agreed. In terms of MALA system will help to improve the HCPs' performance in newborn care, 13 mothers (61.9%) agreed, 7 (33.3%) strongly agreed and the remaining mother (4.8%) shared neutral agreement. In terms of whether mothers would recommend MALA system to other mothers, 33.3% (n = 7) of the participating mothers responded 'neutral', 'agree' and 'strongly agree', respectively.

## Discussion

This study evaluated the mothers' acceptability of a novel technology, which included video and audio recordings of newborn resuscitation, through a survey of 21 mothers whose newborn required neonatal resuscitation. Most of the mothers were comfortable using video recording of the infants and trusted that the information of their infants will be kept confidential. Mothers think that the video and audio recording during resuscitation will neither cause harm nor will compromise the care of infants. Most mothers think that MALA system can help improve HCPs' performance.

Over the past several decades, emerging technology in the delivery room and newborn care has played a significant role in improving clinical outcomes of high-risk newborns [35]. In particular, video recording technology has been used to record newborn resuscitations for data documentation, performance audit and education purposes to identify frequent deviations from the international guidelines and to train and improve HCPs' performances on newborn resuscitation [36,37]. However, video recording technology has only been used retrospectively,

and despite benefits, video recording of newborn resuscitation is not adopted widely. Therefore, real-time use of video and its utilization in supporting and guiding newborn resuscitation care has been considered to be a key area for development [35].

Use of video technology in newborn resuscitation requires acceptance of parents and HCPs as it involves video recording of newborns in a most critical situation. However, limited data are available on how parents perceive video technology. In a previous study, parents perceived video recording and other monitoring technologies used during resuscitation of their babies as an important advancement in neonatal care technology [38,39]. In Australia, 96% of parents were satisfied with the provisions put in place to videotape complicated newborn resuscitation [40]. New technologies can improve parents' emotional well-being by allowing them to feel more connected with their babies during resuscitation, when they would otherwise be unable to be with them [41]. It has been demonstrated that bedside resuscitation improves communication between clinicians and parents and increases acceptability of neonatal resuscitation procedures [31].

In our study, mothers felt comfortable with the newborn care being video recorded and were comfortable with someone using a tablet when observing their infant care. However, when mothers were stratified by their education level, mothers with higher education level tended to show higher acceptability on the newborn care being video recorded and someone using a tablet when observing the newborn care compared with those with lower education level. There is ample evidence that socioeconomic status, particularly education level, have a significant impact on health outcomes and health-related behaviors [42–43]. Moreover, parental education beyond 12 years of schooling is shown to be associated with increases in family health care spending and with reductions in the likelihood of adverse health conditions [44].

Information and consent to video and audio recording are required when MALA system is implemented, therefore adequate information should be provided to the parents, particularly for those with lower education level, for their acceptance of the MALA system. It has been observed that parents would be reluctant to use video technology in neonatal units if it had a negative impact on their child's care, and if it had the potential to negatively modify the behavior of HCPs [38]. Mothers participating in our study perceived that this study (MALA system) will help to improve the health worker's performance in newborn care. Similar to our findings, a study conducted by Yeo *et al.* in Singapore reported that parents were satisfied with use of technologies during resuscitation procedure since it gave them the impression that their baby was being constantly monitored during care [45]. Another study by Kerr *et al.* in Scotland reported that parents perceive such interventions to improve the field of neonatology and contribute to better newborn health outcomes by assisting in understanding of newborn behavior and staying prepared for the next steps of care [41].

While responses from participating mothers showed high indices of acceptance toward the use of MALA technology during newborn resuscitation and strongly felt that the technology helped to improve the HCPs' performance in newborn care, one third of the participating mothers showed neutral/no opinion toward recommending other mothers to participate this study. Since this study was quantitative survey, we do not exactly know the reason why many mothers were hesitant to recommend this novel technology to other mothers. One possible explanation is that mothers may not have enough knowledge about this technology at the level they could explain about the MALA system and recommend to other mothers. This warrants further studies, including qualitative study.

Previous studies have examined the ethical concerns on the use of video recording during resuscitation as it breaches anonymity of the infant receiving care. In a study conducted by den Boer MC *et. al.* in Netherlands, some parents were concerned about the video recordings' privacy, while others saw the recordings of their babies' resuscitation at birth as valuable

documentation about their births for the future and even requested copies [46]. In this study, all women were consented to study at the time of admission and were informed on the anonymity of the recording if such event occurred. Another study by Hawkes *et al.* showed that most of the parents were confident in the security of the systems including the privacy concerns [47].

The results of this study must be taken with the following considerations. First, inclusion of participating mothers was done on a voluntary basis, which may bias the results by recruiting participants with strong positive views regarding video recording. To mitigate the bias in selection process, we compared mothers who participated vs those who did not in terms of their socio-demographic characteristics and found no difference between the group except for education. (S2 Table). Second, our results could not be generalized to other institutions or countries since this was a single-site study with a small sample size and parents are often not allowed to at the bedside during resuscitation or observe resuscitation procedures in low-resource settings. Therefore, study results and conclusion should be interpreted carefully. Third, sample size was not adequate to assess the difference in acceptability by the maternal education. A survey with adequate sample size, will be required to come to the conclusion that there was difference in acceptability rate by maternal education. Fourth, the study assessed the acceptability of video recording with display of timing of ventilation in the resuscitation monitor, which was accepted by the mother. However, the MALA system is further being developed with audio feedback to health care provider on steps of resuscitation. This further improvised MALA system needs to be assessed for acceptability of the parents. Lastly, since this study evaluated mothers' acceptability of the first phase of the MALA system without real-time automated feedback, further studies evaluating acceptability of the complete MALA system are warranted.

## Conclusion

Mothers were hesitant recommending the technology to other mothers, which provides a valuable information to the technology development team and research team. There is a need to further refine the technology through a co-design process with the mothers and caregivers, so that it is acceptable to mothers. Though studies from high income setting showed parent accepting video recording of resuscitation events where the parent's health literacy is different than the mothers from the study site. Thus, valuable information from the study provides two major learnings for this research and other research which recording sensitive as well as critical data point on immediate newborn care. First, mothers should be engaged in the development of the technology, so that they accept the ergonomics as well as the intent of the technology and second, there is a need to improve the maternal health literacy of the video and audio recording of the immediate newborn care. Further, mothers positively accepted the novel technology that uses video and audio recordings during newborn resuscitation and felt that the technology helped to improve the HCPs' performance in newborn care. This provides further support to the MALA system, a deep learning software, being a promising tool for improving neonatal resuscitation quality in low-income settings such as Nepal.

## Supporting information

**S1 Table. Reliability assessment of seven items using Cronbach's alpha.**
(DOCX)

**S2 Table. Comparison of the socio-demographic characteristics between women who consented and those who did not consent.**
(DOCX)

**S3 Table. Acceptability by age (dichotomized based on median age of 25).**
(DOCX)

**S4 Table. Acceptability by parity (first baby vs. 2 or more babies).**
(DOCX)

**S5 Table. Acceptability by education level (up to higher secondary level vs. bachelor's level or higher).**
(DOCX)

**S6 Table. Acceptability by having a computer at home (yes vs. no).**
(DOCX)

**S1 Data. Dataset used for analysis.**
(CSV)

## Acknowledgments

We would like to thank technical advisory group of the project. We would also like to thank all the data collectors, health workers, and parents involved in the study.

## Author Contributions

**Conceptualization:** Øystein Gomo, Fredrik Ahlsson, Yuba Nidhi Basula, Sunil Mani Pokharel, Helge Myklebust, Ashish KC.

**Data curation:** So Yeon Joyce Kong, Omkar Basnet, Anna Axelin, Helge Myklebust.

**Formal analysis:** So Yeon Joyce Kong, Ankit Acharya, Helge Myklebust.

**Funding acquisition:** Solveig Haukås Haaland.

**Investigation:** So Yeon Joyce Kong, Ankit Acharya, Omkar Basnet, Solveig Haukås Haaland, Rejina Gurung, Øystein Gomo, Fredrik Ahlsson, Øyvind Meinich-Bache, Anna Axelin, Helge Myklebust.

**Methodology:** So Yeon Joyce Kong, Omkar Basnet, Solveig Haukås Haaland, Rejina Gurung, Øystein Gomo, Øyvind Meinich-Bache, Sunil Mani Pokharel, Helge Myklebust, Ashish KC.

**Project administration:** So Yeon Joyce Kong, Omkar Basnet, Solveig Haukås Haaland, Rejina Gurung, Øystein Gomo, Yuba Nidhi Basula, Sunil Mani Pokharel, Hira Subedi, Ashish KC.

**Resources:** Rejina Gurung, Øystein Gomo, Øyvind Meinich-Bache.

**Software:** Omkar Basnet, Solveig Haukås Haaland, Øyvind Meinich-Bache, Anna Axelin, Hira Subedi, Helge Myklebust.

**Supervision:** Rejina Gurung, Fredrik Ahlsson, Yuba Nidhi Basula, Sunil Mani Pokharel, Hira Subedi, Ashish KC.

**Validation:** So Yeon Joyce Kong, Rejina Gurung, Fredrik Ahlsson, Øyvind Meinich-Bache, Anna Axelin, Yuba Nidhi Basula, Sunil Mani Pokharel, Hira Subedi, Helge Myklebust.

**Visualization:** Ankit Acharya, Øystein Gomo, Anna Axelin.

**Writing – original draft:** Ankit Acharya.

**Writing – review & editing:** So Yeon Joyce Kong, Omkar Basnet, Solveig Haukås Haaland, Rejina Gurung, Øystein Gomo, Fredrik Ahlsson, Øyvind Meinich-Bache, Anna Axelin, Yuba Nidhi Basula, Sunil Mani Pokharel, Hira Subedi, Helge Myklebust, Ashish KC.

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
