## [Decision Letter · Decision Letter 0]

22 Sep 2023

PDIG-D-23-00193

Mothers’ acceptability of using novel technology with video and audio recording during newborn resuscitation: A cross-sectional survey

PLOS Digital Health

Dear Dr. Ashish.

Thank you for submitting your manuscript to PLOS Digital Health. After careful consideration, we feel that it has merit but does not fully meet PLOS Digital Health's publication criteria as it currently stands. Therefore, we invite you to submit a revised version of the manuscript that addresses the points raised during the review process.

Please submit your revised manuscript within 60 days Nov 21 2023 11:59PM. If you will need more time than this to complete your revisions, please reply to this message or contact the journal office at digitalhealth@plos.org. Please include the following items when submitting your revised manuscript:

We look forward to receiving your revised manuscript.

Kind regards,

Kamrul Hasan, Ph.D.

Guest Editor

PLOS Digital Health

Journal Requirements:

2. In the online submission form, you indicated that "Data will be made available on request". All PLOS journals now require all data underlying the findings described in their manuscript to be freely available to other researchers, either 1. In a public repository, 2. Within the manuscript itself, or 3. Uploaded as supplementary information.

Additional Editor Comments (if provided):

Reviewers' comments:

Reviewer's Responses to Questions

**Comments to the Author**

1. Does this manuscript meet PLOS Digital Health’s publication criteria? Is the manuscript technically sound, and do the data support the conclusions? The manuscript must describe methodologically and ethically rigorous research with conclusions that are appropriately drawn based on the data presented.

Reviewer #1: Yes

Reviewer #2: Yes

Reviewer #3: Partly

2. Has the statistical analysis been performed appropriately and rigorously?

Reviewer #1: Yes

Reviewer #2: Yes

Reviewer #3: Yes

3. Have the authors made all data underlying the findings in their manuscript fully available (please refer to the Data Availability Statement at the start of the manuscript PDF file)?

Reviewer #1: Yes

Reviewer #2: Yes

Reviewer #3: Yes

4. Is the manuscript presented in an intelligible fashion and written in standard English?

Reviewer #1: Yes

Reviewer #2: Yes

Reviewer #3: Yes

5. Review Comments to the Author

Reviewer #1: Studied sample size is very low, so any conclusion regarding the mother's acceptability for the video recording of their newborn's resuscitation couldn't be drawn.

Also, the studied mothers were selected on Voluntary basis so acceptability of the technology in mothers has not been properly established. The study needs to be evaluated, also in mothers having elementary level of education, as per the findings mothers with lower education level were not so comfortable with the technology, so proper survey needs to be done.

Reviewer #2: Interesting paper with a solid methodological basis and reporting of findings. It would benefit from proofreading and some sections in the results can be condensed to as it is reported in the tables.

Reviewer #3: First of all, thank you for doing work on trying to improve newborn resuscitation. I believe this is a valuable research area and I hope that you continue working to develop MALA. However, I believe the current state of the project is not ready for publication given the type of analysis performed in this study.

106-107: There may be several other reasons for MALA to not be accepted by mothers, such as perceived lack of skill from HCPs due to reliance on the technology, lack of trust in technology to guide life saving actions, etc. This paper primarily only tackles the recording aspect, which the authors have already pointed out has been investigated before with varying results (315 - 322). Authors also mention this limitation on 328-330.

229-230: Questions are a little leading. For example, for “I was comfortable with the baby’s care being video recorded” could have been “How did you feel about the baby’s care being video recorded” and have “uncomfortable”, “slightly uncomfortable”, "neutral”, “fairly comfortable”, comfortable”. This makes me question the reliability of the results.

310-312: The lack of information about why mothers were hesitant to recommend this novel technology to other mothers makes the contribution of this paper weaker. I do not see how this submission could influence the future direction of the development of the technology or how impact how to more effectively get mothers to accept the technology. Given that other publications that the authors point out have also looked into the acceptance of video recording infants, the contribution of this paper seems limited.

323-325: The bias from only having data for individuals who participated in the study is a big concern, especially considering that there were 3 deliveries where the mothers did not consent to participate in the survey and their positions are unrepresented here.

6. PLOS authors have the option to publish the peer review history of their article (what does this mean?). If published, this will include your full peer review and any attached files.

**Do you want your identity to be public for this peer review?** For information about this choice, including consent withdrawal, please see our Privacy Policy.

Reviewer #1: No

Reviewer #2: Yes: Tigest Tamrat

Reviewer #3: No

---

## [Decision Letter · Decision Letter 1]

19 Feb 2024

Mothers’ acceptability of using novel technology with video and audio recording during newborn resuscitation: A cross-sectional survey

PDIG-D-23-00193R1

Dear Dr. KC,

We are pleased to inform you that your manuscript 'Mothers’ acceptability of using novel technology with video and audio recording during newborn resuscitation: A cross-sectional survey' has been provisionally accepted for publication in PLOS Digital Health.

Best regards,

Laura Sbaffi, PhD, MA, MSc

Section Editor

PLOS Digital Health

Reviewer Comments (if any, and for reference):

Reviewer's Responses to Questions

**Comments to the Author**

1. If the authors have adequately addressed your comments raised in a previous round of review and you feel that this manuscript is now acceptable for publication, you may indicate that here to bypass the “Comments to the Author” section, enter your conflict of interest statement in the “Confidential to Editor” section, and submit your "Accept" recommendation.

Reviewer #1: All comments have been addressed

2. Does this manuscript meet PLOS Digital Health’s publication criteria? Is the manuscript technically sound, and do the data support the conclusions? The manuscript must describe methodologically and ethically rigorous research with conclusions that are appropriately drawn based on the data presented.

Reviewer #1: Yes

3. Has the statistical analysis been performed appropriately and rigorously?

Reviewer #1: Yes

4. Have the authors made all data underlying the findings in their manuscript fully available (please refer to the Data Availability Statement at the start of the manuscript PDF file)?

Reviewer #1: Yes

5. Is the manuscript presented in an intelligible fashion and written in standard English?

Reviewer #1: Yes

6. Review Comments to the Author

Reviewer #1: All the raised points are answered appropriately, also the required changes have been made accordingly, but the study needs to be considered as a reference for a future study, to claim the acceptance of the MALA technology on a large scale.

7. PLOS authors have the option to publish the peer review history of their article (what does this mean?). If published, this will include your full peer review and any attached files.

**Do you want your identity to be public for this peer review?** For information about this choice, including consent withdrawal, please see our Privacy Policy.

Reviewer #1: No
